# Drivers of menstrual material disposal and washing practices: A systematic review

**Hannah Jayne Robinson**[1]*, **Dani Jennifer Barrington**[1,2]

**1** University of Leeds, Leeds, West Yorkshire, United Kingdom, **2** The University of Western Australia, Crawley, Western Australia, Australia

* cn16hjr@leeds.ac.uk

**Data Availability Statement:** This manuscript made use of secondary data in the form of publications reporting on menstrual health. Table 2, Supplementary 2 and the Reference list provide the

## Abstract

### Background

Disposal and washing facilities and services for menstrual materials are often designed based upon technical specifications rather than an in-depth understanding of what drives peoples' choices of practices.

### Objectives and data sources

This systematic review identified and summarised the main behavioural drivers pertaining to the choice of disposal and washing practices of menstrual materials through the thematic content analysis and study appraisal of 82 publications (80 studies) on menstrual health and hygiene published since 1999, reporting the outcomes of primary research across 26 countries.

### Results

Disposal and washing behaviours are primarily driven by the physical state of sanitation facilities; however, this is intrinsically linked to taboos surrounding and knowledge of menstruation.

### Implications

Using reasons given for disposal and washing practices by menstruators or those who know them well, or inferred by authors of the reviewed studies, we identify the key considerations needed to design facilities and services which best suit the desired behaviours of both planners and those who menstruate.

### Inclusivity

The term menstruators is used throughout to encompass all those mentioned in the studies reviewed (girls and women); although no studies explicitly stated including non-binary or transgender participants, this review uses inclusive language that represents the spectrum of genders that may experience menstruation.

details of all publications included in this systematic review.

**Funding:** The lead author of this paper is supported by the Engineering and Physical Sciences Research Council Grant number EP/S022066/1.

**Competing interests:** The authors have declared that no competing interests exist.

## Registration

The review protocol is registered on PROSPERO: 42019140029.

## Introduction

Menstrual health and hygiene (MHH) are an integral part of public health, recognized by an increase in research on this topic in the past decade [e.g. 1,2], and the recent definition of menstrual health [3]. Within the MHH space there has been a lot of research into the provision of menstrual materials, and subsequent interventions that provide those who menstruate with both reusable and disposable materials [4–6]. However, there has been less research into what happens once these materials have been used; the full lifecycle of these materials has often not been documented. Understanding the full lifecycle of menstrual materials is especially important for those designing the infrastructure of water, sanitation and hygiene (WASH) programs, including, but not limited to, toilets, bathing facilities, washing and drying facilities, incinerators, and solid waste management services. Water, sanitation and hygiene facilities need to be technically and socially appropriate to allow people to change and dispose of menstrual materials safely for them, the associated infrastructural systems and the environment. Disposal choices directly affect the functioning of sanitation systems; if materials are discarded in toilets or pit latrines, they can create blockages which reduce functionality of a system [7]. Disposal and washing practices can also have adverse health effects on users, for example, if there are no spaces for drying reusable materials, it is possible that infections could manifest if materials are used before they are dried properly [8].

Currently, the drivers behind menstruators' choice of disposal and washing practices are often not documented and rarely considered when WASH facilities are designed. For example, although it alludes to the need for 'cultural considerations' when designing facilities, the International Standard on Non-Sewered Sanitation requires technology developers to provide users with instructions on how to dispose of their menstrual materials so as to protect the functioning of the technology, but this does not necessarily consider the reasons why a user may choose to flush materials despite knowing that it may harm the infrastructure [9]. This views users through a deficit lens [10], it assumes that if only they knew better they would change their behaviour. But WASH behaviours are not always driven by possessing the appropriate 'knowledge', for example, people may have been taught that it is unsafe to practice open defecation but choose to do so for reasons of convenience, pride and mental well-being [11]. Such disconnects between 'knowledge' and 'action' are why the field of WASH behaviour change scholarship exists [12].

Two systematic reviews have been published compiling the methods of menstrual disposal used in low and middle income countries [13,14], but neither thoroughly investigates why users practice these behaviours. A recent critical review of 'unflushable' objects entering waterborne sewerage highlights the lack of research into the drivers behind user decisions to dispose of non-biodegradable materials in toilets around the world [15]. No reviews have been published on menstrual material washing practices.

To better design WASH systems that meet the needs of those who menstruate it is important to understand what disposal and washing practices are currently used, but more importantly, why: if WASH professionals can understand what drives washing and disposal methods in different contexts, they can design technically robust systems that those who menstruate want to use. To address this, we systematically investigated the extant peer-reviewed literature on menstrual disposal and washing practices so as to determine: 1) What drives the behaviours

of those who menstruate when deciding on a method of disposal or washing of used menstrual products?; 2) Are there differences in behavioural drivers and practices between low, middle and high income economies?; 3) If MHH programming is to be socially, environmentally, economically and technically sustainable, how does it need to engage with the drivers of behaviour around disposal and washing?

## Method

The review protocol is registered on PROSPERO: 42019140029 (https://www.crd.york.ac.uk/prospero/display_record.php?RecordID=140029) and is reported according to PRISMA guidance [16 and S1 PRISMA Checklist].

### Search strategy

A systematic search of peer-reviewed literature was conducted according to the PRISMA guidelines [17] (Fig 1) and included documents published since 1999, to ensure that findings were relevant to current disposal and washing practices.

Topics to be searched for in the documents included: solid waste disposal; menstrual waste disposal; health effects regarding disposal and material usage, and factors affecting disposal routes (Table 1). Searching was undertaken in eight databases in June 2019 and updated in June 2021.

This yielded the following results for initial screening: Scopus (8,367), Web of Science (5,050), EBSCO (consisting of CINAHL, GreenFILE, Social Work Abstracts, Child Development & Adolescent Studies) (1,681), MEDLINE (5,005) and Proquest Dissertations and theses (440). We also hand-searched the bibliographies of the two existing reviews on low/middle income country menstrual material disposal methods [13,14].

After removing duplicates, titles and/or abstracts of 14,198 publications were screened against Criteria 1 (primarily about menstruation or sexual and reproductive health), followed

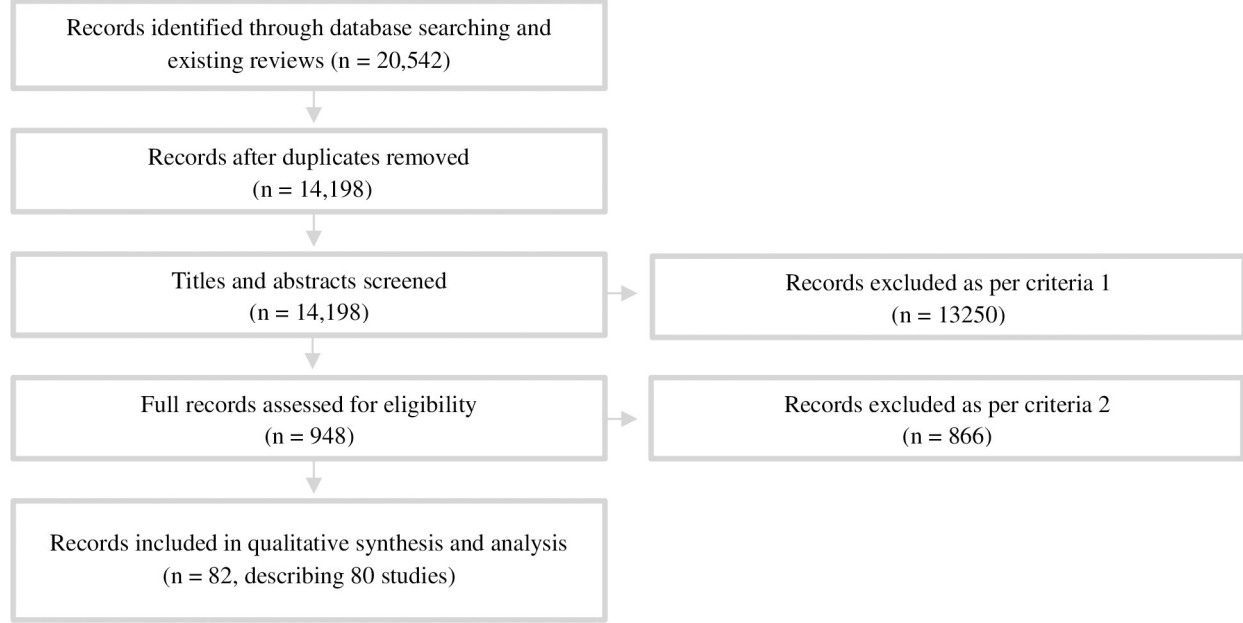

**Fig 1. Inclusion and exclusion flowchart for systematic review [17].** Criteria 1: Primarily about menstruation or sexual and reproductive health; Criteria 2: Published post-1999 and discusses behaviours post-1999, in English or has an English translation available, discusses menstrual material disposal, washing, drying and/or reuse and gives reasons for these behaviours.

**Table 1. Scopus search strategy.**

| Search 1: Menstruation | (menstru* or menarche).ab,kw,ti. |
|---|---|
| Search 2: Disposal or washing | (wash* OR dispos* OR dri* OR dry* OR recycl* OR reus* OR process* OR waste* or "used product*" OR threw* OR throw* OR rubbish OR garbage OR landfill OR bin OR hide OR bury* OR buried OR burn* OR hygien*).ab,kw,ti. |
| Search 3: Final | 1 AND 2 |

ab = abstract, kw = keywords, ti = title.

by the full-text screening of 948 publications against Criteria 2 (published post-1999 and discusses behaviours post-1999; in English or has an English translation available; discusses menstrual material disposal, washing, drying and/or reuse and gives reasons for these behaviours). This led to the inclusion of 82 publications (80 studies) (Table 2 provides a summary of the included studies, for full details of each see S1 Table). Grey literature was not searched as the themes which arose from the peer-reviewed publications reached saturation.

## Quality appraisal

Included studies were assessed on their level of trustworthiness and relevance by adapting the method of Rees et al [100] to be applicable to both quantitative and qualitative MHH studies. The assessment of trustworthiness was dependant on the sampling, data collection and analysis, and interpretation of data; this yielded high, medium, and low ratings. Relevance related to the proportion of the paper dedicated to menstrual material disposal or washing, whether confidentiality had been assured and appropriate consent obtained, and who gave the reasons for behaviours; this yielded high, medium, and low ratings (for full details of the quality appraisal of each study, see S2 Table). We considered studies which took a social constructivist approach to understand drivers of menstrual disposal and washing, evidenced by the experiences of menstruators, to be of higher relevance than those which presented less direct evidence and tended along positivist epistemological lines. Considering the trustworthiness and relevance of studies allowed us to weight themes more heavily where they were results of robust research, particularly where the reasons for disposal and washing behaviours were given by those who menstruate themselves.

## Thematic content analysis

After reviewing the full texts of each publication, deductive and inductive coding was undertaken using NVivo 12 [101]:

1. Examples of practices were deductively coded as "intentional disposal" or "washing drying or reuse".

2. Axial coding was conducted to identify the drivers behind each practice (see Table 3 for codebook and coding frequency). This led to an understanding of the influence of physical and social drivers on menstrual material disposal and washing practices and how these must be considered by those planning WASH facilities.

3. Publications were also classified demographically in order to understand the study context (Tables 2 and S1).

A conceptual model was developed to understand the breadth of reasoning behind disposal and washing practices.

**Table 2. Included studies.**

| Citation | Country | Economic status | Population | Sample Size |
|---|---|---|---|---|
| Abera, 2004 [18] | Ethiopia | Low | Girls in school*, school staff | 863 questionnaires (female, grades 9–10 (≈ aged 14–16), across 8 schools), 4 focus group discussions (8 students in each), and key informant interviews with school authorities (number unspecified) |
| Ahmmed et al, 2021 [19] | Bangladesh | Lower-middle | Adolescent girls, Women, Birth Attendants and Medicine Vendors | 89 married women (reproductive age), 42 adolescent girls (aged 14–18), 18 elderly women, 3 traditional birth attendants, 3 medicine vendors |
| Alda-Vidal and Browne, 2021 [20] | Malawi | Low | Women | 40 Women (age unspecified), 13 sanitation workers and 15 external MHM actors |
| Alexander et al, 2014 [21] | Kenya | Lower-middle | School staff | 62 Headteachers (age unspecified) |
| Asimah et al, 2017 [22] | Ghana | Lower-middle | Girls in school*, guardians | 319 pupils (aged 10–19, with 229 females, 90 males across 15 schools), and 333 household heads (241 males, 92 females) |
| Averbach, et al, 2009 [23] | Zimbabwe | Lower-middle | Women | 43 women (aged 18–45) |
| Behera et al, 2015 [24] | India | Lower-middle | Adolescent girls** | 32 adolescent girls (female, aged 14–15) |
| Bhattacharjee, 2019 [25] | India | Lower-middle | Women and Adolescent Girls | 84 Women and adolescent girls (aged 15–50, across 3 villages) |
| Caruso et al, 2017 [26] | India | Lower-middle | Women | 69 women (for interviews, aged 18–75), and 46 women (for discussions, aged 18–70) |
| Caruso et al, 2014 [27] | Kenya | Lower-middle | Girls in school*, school staff | 36 students (female, aged 11–17, across 3 primary schools for focus groups), 6 students (selected from the focus group discussions, for in-depth interviews), 2 teachers (for in-depth interviews) |
| Chakravarthy et al, 2019 [28] *(Paper uses 3 studies– 1 available report and 2 unpublished documents)* | India | Lower-middle | Women & girls, Government officials | Unspecified number of adolescent girls (aged 10–19) women (aged 20–49) and 20 government officials. Breakdown of participants not specified. |
| Chinyama et al, 2019 [29] | Zambia | Lower-middle | Girls in school*, school staff, guardians | 64 students (aged 14–18, 48 female, 16 male, for 8 focus group discussions), 12 students (aged 14–18, female, for in-depth interviews), 7 teachers (for key informant interviews), (all across 6 schools), 7 guardians (for key informant interviews), and 11 leaders (both male and female (for key informant interviews) |
| Chothe et al, 2014 [30] | India | Lower-middle | Girls in school* | 381 students (female, aged 9–13) |
| Connolly and Sommer, 2013 [31] | Cambodia | Lower-middle | Adolescent girls**, school staff, guardians | 146 adolescent girls (female, aged 16–19, mix of in and out of school), and 15 parents/ teachers |
| Coswosk et al, 2019 [32] | Brazil | Upper-middle | Girls in school*, school staff | School principal and vice-principal, 39 students (female and male, aged 13–17) |
| Crankshaw et al, 2020 [33] | South Africa | Upper-middle | Girls in school, Boys in School, School staff, Mothers of Girls in School | 505+ students (across 10 schools), 8 teachers, 9 mothers of students, (Breakdown of school students not specified) |

*(Continued)*

**Table 2.** (Continued)

| Citation | Country | Economic status | Population | Sample Size |
|---|---|---|---|---|
| Crichton et al, 2013 [34] | Kenya | Lower-middle | Adolescent girls** | 87 students (aged 12–17), 69 women, 5 teachers, 1 nurse |
| Crofts and Fisher, 2011 and Crofts and Fisher, 2012 [35,36] | Uganda | Low | Girls in school*, school staff, business leaders | 134 students (female, aged 13–20, for participatory activities and FDGs), 9 business leaders, 12 school staff |
| Daniels, 2016 [37] | Cambodia | Lower-middle | Adolescent girls**, adolescent boys, women, men, school staff | 165 participants (for interviews), 181 participants (for focus group discussions), including girls, boys, mothers, fathers, and teachers. Breakdown of participants not specified. |
| Dhingra et al, 2009 [38] | India | Lower-middle | Adolescent girls** | 200 girls (aged 13–15) |
| Dolan et al, 2014 [39] | Ghana | Lower-middle | Girls in school*, parents, school staff | 99 girls (age unspecified, for interviews), 136 girls (age unspecified, including dropouts, for focus group discussions), 246 parents, 12 school staff (for key informant interviews), 156 school staff (for focus group discussions) |
| Ellis et al, 2016 [40] | Philippines | Lower-middle | Girls in school* | 79 students (female, aged 11–18, across 3 schools in urban Manilla, and 10 rural schools) |
| Enoch at el, 2020 [41] | Ghana | Lower-middle | Adolescent girls | 18 adolescent girls (aged 12–19, with visual, hearing or physical disabilities (6 girls for each disability)) |
| Garikipati and Boudot, 2017 [42] | India | Lower-middle | Adolescent girls**and women | 150 women and adolescent girls (aged 15–49, from 3 slum locations) |
| George and Leena, 2020 [43] | India | Lower-middle | Women | 22 women (aged 25–49) |
| Girod et al, 2017 [44] | Kenya | Lower-middle | Girls in school*, school staff | 51 students (approximately–number of students not explicitly stated, female, grades 6–8 ($\approx$ aged 11–14), across 6 different primary schools) and 6 Headteachers |
| Gultie et al, 2014 [45] | Ethiopia | Low | Girls in school* | 492 students (female, grades 9–12, aged 13–21+) |
| Habtegiorgis et al, 2021 [46] | Ethiopia | Low | Girls in School | 536 students (female, aged 13–19, across 5 schools (3 public, 2 private, 457:79) |
| Hawkins et al, 2019 [47] | UK | High | Women | 10 women (female, aged 18–30) |
| Hennegan et al, 2020 [48] | Uganda | Low | Women | 35 Women (female, aged 18–35) |
| Hennegan and Sol, 2020 [49] | Bangladesh | Lower-middle | Girls in school | 1359 students (female, aged 10–16, across 149 schools, (approximately 9 students per school)) |
| Hennegan et al, 2017 [50] | Uganda | Lower-middle | Girls in school* | 27 students (female, aged 12–17, across 8 schools) |
| Hennegan et al, 2016 [51] | Uganda | Lower-middle | Girls in school* | 205 students (female, aged 10–19, across 8 schools) |
| Htun et al, 2021 [52] | Myanmar | Lower-middle | Adolescent girls | 410 adolescent girls (aged 9–15, across 38 villages) |

(*Continued*)

**Table 2.** (*Continued*)

| Citation | Country | Economic status | Population | Sample Size |
|---|---|---|---|---|
| Jahan et al, 2020 [53] | Bangladesh | Lower-middle | Girls in school | Pre-intervention Period (PrIP):168 students and 17 school staff // Intervention Design Period (IDP): 139 students and 12 school staff // Post-intervention Period (PIP): 100 students and 20 school staff // 468 individuals, including students (419), teachers (21), and janitors (28) (All students aged 12–16) |
| Kambala et al, 2020 [54] | Malawi | Low | Women, girls in school, school staff, community leaders, community health workers, and MHM service providers | 80 students (female, aged 10–18), 61 women, 12 school staff, 6 community leaders, 8 community health workers, and 9 MHM service providers |
| Karibu et al, 2019 [55] | Nigeria | Lower-middle | Adolescent girls** | 492 adolescent girls (aged 10–19, covering both those in and out of school) |
| Kemigisha et al, 2020 [56] | Uganda | Low (*refugee settlement*) | Adolescent girls | 28 adolescent girls (aged 13–19) |
| Kohler et al, 2019 [57] | India, Uganda | Lower-middle | Women, and men (inpatients and healthcare staff) | 50 Indian participants and 40 Ugandan participants (across 4 hospitals, for workshops, interviewees selected from this sample). Both samples included inpatients and staff. |
| Kumbeni et al, 2020 [58] | Ghana | Lower-middle | Girls in school | 730 students (female, aged 10–19, across 15 schools) |
| Lahme et al, 2018 [59] | Zambia | Lower-middle | Girls in school* | 51 students (female, aged 13–20, across 3 schools) |
| MacRae et al, 2019 [60] | India | Lower-middle | Women | 114 Women (across 12 communities– 39 unmarried women, 12 recently married women, 38 married women, 25 older women, age unspecified) |
| Mason et al, 2013 [61] | Kenya | Lower-middle | Girls in school* | 120 Students (female, aged 14–16, cross 6 schools) |
| Maulingin-Gumbaketi et al, 2021 [62] | Papua New Guinea | Low-middle | Women | 98 women (aged 13–45+, across 4 provinces) |
| McHenga et al, 2020 [63] | Malawi | Low | Girls in school and school staff | 228 students (female, aged 11–22), 22 school staff (Head Teachers and Senior female teachers) |
| Miiro et al, 2018 [64] | Uganda | Lower-middle | Girls in school*, boys in school, school staff, Municipality officials, parents | 562 Students (352 female and 210 male, aged 13–18, across 4 schools), 11 teachers, 2 municipality officials (Ministry of Education and the Ministry of Health), 10 parents |
| Mohamed et al, 2018 [65] | Fiji, Papua New Guinea, Solomon Islands | Lower-middle | Women & girls, men, school staff, community members (including vendors, employers, health workers, community leaders and vulnerable women) | 54 girls in school (aged 13–26), 43 adolescent girls (aged 13–29), 118 women (aged 19–61), 51 men (aged 23–70), 8 school staff, and 34 community members |
| Mohammed and Larsen-Reindorf, 2020 and Mohammed et al, 2020 [66,67] | Ghana | Lower-middle | Girls in school, boys in school and 5 school staff | 280 Students (250 female, aged 10–19, across 5 schools; 30 male, across 3 schools) and 5 head teachers |
| Mumtaz et al, 2019 [68] | Pakistan | Lower-middle | Girls in school, women, school staff, care providers, local religious leaders and a scholar | 312 students (female, aged 16–19 years), 15 mothers, 11 female school teachers, 9 health care providers, 5 local religious leaders and 1 scholar |

(*Continued*)

**Table 2.** (Continued)

| Citation | Country | Economic status | Population | Sample Size |
|---|---|---|---|---|
| Muralidharan, 2019 [69] | India | Lower-middle | Women & girls | Up to 72 adolescent girls (aged 15–24), and their mothers (total of number of participants not stated) |
| Nalugya et al, 2020 [70] | Uganda | Low | Girls in school, parents, school staff | 450 Students (baseline: 232 female and 218 male, aged 13–21, across 2 schools) 369 Students (endline: 188 female and 181 male, aged 13–21, across 2 schools), 10 parents, 10 teachers |
| Ndlovu and Bhala, 2016 [71] | Zimbabwe | Lower-middle | Women, NGOs, Public Sector, Religious institutions | 40 women, 30 key informants (15 males and 5 females, including public sector departments, churches and NGOs) |
| Oche et al, 2012 [72] | Nigeria | Lower-middle | Adolescent girls** | 122 adolescent girls (aged 15–20, across 4 schools) |
| Parker et al, 2014 [73] | Uganda | Lower-middle, and Displacement Camp (in and out of displacement camps) | Girls in school*, women, school staff, health workers | Up to 240 students (aged 9–20, across 14 schools), 8 senior/head teachers, 9 health workers, up to 75 women (across 4 villages), up to 450 women (across 13 IDP settings) (Total of number of participants not stated) |
| Rajagopal and Mathur, 2017 [74] | India | Lower-middle | Adolescent girls** | 270 adolescent girls (130 school-going, 140 non-school-going, aged 10–20, across 5 schools) |
| Rajaraman et al, 2013 [75] | India | Lower-middle | Women | 48 women (age unspecified) |
| Ramathuba, 2015 [76] | South Africa | Upper-middle | Girls in school* | 273 students (female, aged 14–19, across 6 schools) |
| Rastogi et al, 2019 [77] | India | Lower-middle | Girls in school*, parents, school staff | 187 students (female, aged 13–15, across 4 schools), parents and school staff. Total of number of participants not stated. |
| Rheinländer et al, 2019 [78] | Ghana | Lower-middle | Girls in school*, school staff | 33 students (female, aged 14–23, across 2 schools), 4 school staff (female) |
| Rizvi and Ali, 2016 [79] | Pakistan | Lower-middle | Adolescent girls** | 20 adolescent girls (aged 13–19, non-school-going) |
| Roxburgh et al, 2020 [80] | Malawi | Low | Women and university staff | 31 women (aged 19–60+) and 2 university staff |
| Schmitt et al, 2017 [81] | Lebanon, Myanmar | Displacement Camp | Women & girls, humanitarian staff | 117 women (aged 19–49), 71 adolescent girls (aged 14–18, 32 for focus group discussions and 39 for participatory mapping), 17 emergency response staff |
| Schmitt et al, 2021 [82] | Bangladesh | Low-middle (refugee settlement) | Women & girls, humanitarian response staff | 47 Adolescent girls and women (aged 15–35), 19 humanitarian response staff |
| Scorgie et al, 2016 [83] | South Africa | Upper-middle | Women | 21 women (aged 18–35, 17 of these completed the photovoice segment, then 7 of these then completed interviews) |
| Shah et al, 2019 [84] | Gambia | Low | Girls in school*, mothers, school staff | 470 students (427 female—aged 11–21, 43 male–aged 15–21), 3 school staff, 5 mothers |
| Sheoran et al, 2020 [85] | India | Lower-middle | Women & girls | 800 Women & girls (aged 14–49) |
| Sivakami et al, 2019 [86] | India | Lower-Middle | Girls in school* | 2564 students (female, aged 12+ (average age 14), across 43 schools) |

(*Continued*)

**Table 2.** (Continued)

| Citation | Country | Economic status | Population | Sample Size |
|---|---|---|---|---|
| Sommer et al, 2015 [87] | Cambodia, Ghana, Ethiopia (This study also draws from a previous study from Tanzania–Sommer, 2009, for comparison purposes, this study is detailed below) | Low and Lower-Middle | Adolescent girls**, school staff, parents, health staff | ≈ 450 adolescent girls (aged 16–19, both in and out of school, across the 3 countries), school staff, parents, health staff (total of number of participants not stated) |
| Sommer, 2009 [88] | Tanzania | Lower-Middle | Adolescent girls** | ≈ 140 adolescent girls (aged 16–19) (Total of number of participants not stated) |
| Sommer et al, 2020 [89] | USA | High (homeless women) | Women, government staff, shelter staff | 22 women (aged 16–62), 3 government staff and 12 shelter staf |
| Tamiru et al, 2015 [90] | Ethiopia, South Sudan, Tanzania, Uganda, Zimbabwe | Low and Lower-Middle | Girls in school*, boys in school, school staff, community members | Total of number of participants not stated (students aged 11+) |
| Tegegne and Sisay, 2014 [91] | Ethiopia | Low | Adolescent girls**, school Staff | At least 595 students (female, aged 10–19), 5 adolescent girls (who had dropped out of school), 4 teachers (all female). Total of number of participants not stated. |
| Trinies et al, 2015 [92] | Mali | Low | Girls in school*, school Staff | 26 students (female, aged 12–17, across 8 schools), 14 school staff (4 female, 10 male, across 8 schools). |
| Umeora and Egwuatu, 2008 [93] | Nigeria | Lower-middle | Women | 1692 women (female, aged 17–56). |
| Visaria and Mishra, 2017 [94] | India | Lower-middle | Adolescent girls** | 585 adolescent girls (aged 12–19, split across the experiment area (406) and a control group (179), spanning rural and urban communities). Total number of participants not explicitly stated. |
| Wardell and Czerwinski, 2001 [95] | USA | High | Women | 33 women (aged 22–27, on active duty or reserve forces for the military) |
| WaterAid Nepal, 2009 [96] | Nepal | Low | Girls in school* | 204 students (female, aged 12–20, across 4 schools) |
| Wilbur et al, 2021 [97] | Nepal | Lower-middle | Women and carers | 20 women and girls (aged 15–24) and 13 carers |
| Wilson et al, 2014 [98] | Kenya | Lower-middle | Girls in School* | 302 students (female, unknown age, across 10 schools) |
| Yeasmin et al, 2017 [99] | Bangladesh | Lower-middle | Women, men, waste emptiers | 43 women, 25 men, 14 children, 5 faecal sludge emptying operators, 4 waste bin emptiers |

*'Girls in school' refers to studies that were specifically conducted in a school environment

**'adolescent girls' refers to participants either not in education, or studies that were set outside the school environment.

# Results

Publications detailed studies in low income economies (17 studies), lower middle income economies (58 studies), upper-middle income economies (5 studies), high income economies (3 studies), and displacement camps (3 studies) (as defined by World Bank in 2020 [102]) (Tables 2 and S1). There was a skew towards girls' (<18 years old) experiences over women's experiences (58 instances vs. 29 instances). There were no studies which detailed the experiences of those who identify as trans-men or gender non-binary.

49 studies were rated as high trustworthiness, 30 medium trustworthiness and 1 low trustworthiness. High trustworthiness papers were characterised by having more than 50 participants, a clear analysis description, and supportive quotes that were clearly distinguishable

**Table 3. Codebook definitions and coding frequency.**

| | Code | | Definition | Studies |
|---|---|---|---|---|
| **Practise (Deductive Codes)** | Intentional Disposal | | Menstruators chose to engage in a certain disposal technique (e.g. throwing into a latrine, field, jungle, canal, or bin; flushing down a toilet; burying them; wrapping them in newspaper, plastic, or paper; or leaving them on the toilet floor) | 56 (70%) |
| | Washing, drying or reuse | | Washing between uses to reuse materials; washing blood of materials for religious/ cultural reasons before disposing; drying in the sun (on roofs/washing lines); or drying inside homes (open-air drying and hiding whilst drying) | 47 (59%) |
| **Reason / Behavioural Driver (Inductive Codes)** | State of Available Facilities *(40 Studies)* | Physical Infrastructure | Does the sanitation facility meet desired physical sanitation needs? | 52 (65%) |
| | | Social Perceptions | Does the sanitation facility meet desired social needs? | 42 (53%) |
| | Knowledge *(11 Studies)* | Lack of knowledge | Menstruators have not been taught how to dispose / wash / dry materials | 14 (18%) |
| | Menstrual Taboos and Social Stigma *(36 studies)* | Cultural Beliefs | General beliefs discouraging / encouraging certain methods of disposal | 28 (35%) |
| | | Embarrassment and Worry | Unpleasant emotions related to doing something considered by others to be wrong or shameful | 35 (44%) |
| | | Fear | Unpleasant emotion caused by the threat of danger, pain or other harmful consequences | 13 (16%) |

between participants. 26 studies were considered of high relevance to this review, 46 medium relevance and 8 low relevance. High relevance studies tended to have a high proportion of the paper discussing findings relevant to disposal and washing of menstrual materials, evidence of behavioural drivers as stated by menstruators themselves, and consent and confidentiality measures stated clearly. Lower relevance studies only briefly mentioned disposal and washing behaviours and/or presented the reasons for behaviours mostly from author inferences, rather than given by menstruators themselves.

In 29 studies, reasons for disposal and washing practise were given by menstruators alone, in 23 reasons were given solely by authors, and in 28 there was a mixture of authors' reasoning and menstruators' reasoning. The reasons given for disposal and washing practices for each study are detailed in S1 Table and a summary of the frequency of reasons is detailed in Table 3. Illustrative examples of the drivers of disposal and washing behaviours are provided in S3 Table.

## Reasoning behind behaviour

When investigating the reasoning behind menstruators' choice to use certain disposal and washing practices, the predominant factor was the availability of appropriate WASH facilities (where 'appropriateness' was defined by users). Of the 80 studies, 56 mentioned that the reason for the disposal or washing practice directly related to the state of the facilities used to manage menstruation, 52 the physical needs of the individual, and 42 the social perceptions of the facility according to individuals. There were 13 instances of menstruators stating lack of knowledge as a reason that affected their behaviour, as they stated they were unsure what was supposed to be done after menstrual materials had been used. Menstrual stigma and taboos were stated in 55 papers as a reason influencing disposal and washing practices.

Where the physical state of WASH infrastructure was mentioned, it related to the quantity of available and physically functional toilets/latrines [22,26,28,32,33,37,40,44,46,58,63,73,75, 76,78,81,86,87,89,91], the design of toilets/latrines [32,37,40,44,48,51,57,63,64,68,71,73,75, 81,83,87,90,97], the quality and availability of running water in and around toilets or latrines [18,20,31,33,37,39,40,45,46,48,49,51,53,54,56–60,62,63,66–68,71,73,77,87,88,91,92,94–96], the

availability of soap for washing [20,33,49,53,54,56–58,60,61,63,64,66,67,73,77,90,91] and the availability of a physically functional disposal mechanism and/or service for used material [18,28,31–37,42,44–48,52,53,57,58,60,62,63,66–68,71,76,77,82,83,86,90,91,95–97].

Social perceptions of appropriate infrastructure were driven by the presence or absence of a private/safe space for managing menstruation [28,34,36,37,39,40,44–46,48,51–54,57,60,63,64,66–69,71,74,76,80,81,83,87,89–92,96], the cleanliness and maintenance of the facility [25–28,31,33,36,37,40,43,44,46,48,53,57,63,66–68,71,74,77,78,87–90,92,96], the time they had available to change, wash or dispose of materials [25,28,33,53,86] and the availability of gender-segregated toilets / latrines [53,58,63,81,87,91,92].

In 14 of the studies menstruators explicitly stated that they had been given no, or limited, advice regarding how to dispose of menstrual materials [21,24,28,30,37,45,64,74,78,81,83]. However, menstruators also stated they chose their method of disposal to limit environmental harm [55], or not cause detrimental harm to infrastructure systems [83] (e.g., disposable pads blocking flush systems).

55 studies indicated that the choice of disposal or washing behaviour was driven by menstrual taboos and social stigma. For example, menstruators that used reusable materials often dried them discreetly to hide them from others, often drying their washed materials inside, sometimes in a hidden corner [38,50,51,55,73,74,84,91–94], under their clothes [60,69,74,76,81], hidden under or within other drying items [25,36,49,60,65,73,96], or generally inside out of view [48,52,58,68]. There were instances of drying menstrual materials outside [46,49,50,55,56,65,73,98] but typically in cases where menstruators believed that there was adequate separation of homes so that neighbours could not see. When using disposable materials, many menstruators wrapped their used menstrual materials in newspaper or polythene bags before disposal, so as to obscure their waste [24,25,32,40,41,43,47,54,60,63,69,74,77,79,81,83,85,97,99]. In addition, some cultures discouraged the use of open disposal, such as bins, due to fears of witchcraft and infertility [19,20,29,30,36,41,42,53,57,60,68,71,72,82,83,87,90,92,93,99], or due to beliefs it is a sin to throw (especially unwashed) materials into a bin [79].

## Discussion

There are three main drivers of menstrual disposal and washing behaviour, and they can be independent or influenced by one another. They can be standalone, for example, there may be no bin within the WASH facility so a menstruator cannot dispose of materials into a receptacle, the menstruator may not have been taught how to dispose of used materials so does not know what to do, or the menstruator may have been taught that they must not incinerate materials lest evil spirits negatively affect their health. However, the reasons for disposal and washing menstrual materials can also be multi-faceted. In some instances all three combine to influence behaviour: for example, a menstruator may not have been taught how to dispose of materials, feel uncomfortable openly discussing disposal options due to menstrual stigma and may not know what facilities are needed or available, or how to change facilities or practices to make them more appropriate.

### Understanding the drivers

The state of WASH facilities was the predominant driver in 57 studies. This included the physical needs and social perceptions of menstruators, but often also incorporated menstrual taboos and social stigma, for example, just because facilities existed, and were technically 'appropriate', did not mean they were used. Crofts and Fisher noted that incinerators had been constructed in five of the 18 schools where interviews took place, so there was a physically functioning menstrual disposal option present, however the actual usage was low [36]. The

incinerators had been built on the opposite side of the school to the toilets, so students did not want to be seen entering the incinerator building, and even when disposal buckets were provided in toilets, there was no management in place to take waste from toilets to the incinerator [36]. So, although there were physically appropriate disposal facilities, due to stigma around being seen holding or disposing of used materials, the disposal facility was not used. The theme of menstruators not wanting to be seen disposing or washing their used materials was driven by two emotions: being embarrassed and/or worried (35 studies), or fearing that they would be seen, and what would happen if they were seen (13 studies).

Embarrassment and worry were direct results of being seen disposing or washing menstrual materials, and often lead to hiding of materials, and secretive behaviours [19,20,25–28,33–35,37,40,44,47–49,53,58,60,62,65,68,73,74,76–81,83–85,92,97,99]. This worry was usually related to having ones' menstrual status exposed (meaning those around the menstruating individual are aware that they are currently experiencing their period). Embarrassment drove disposal and washing behaviours and methods that favoured privacy. This behaviour can be harmful, as if menstruators hide materials when drying them (e.g. under beds or other clothes), they may not be properly dried, and it is possible to contract infections and skin irritation. Menstruators were often aware of the dangers of incomplete drying of reusable products [23,36,54,56,60,71], but felt they have no other alternative due to the stigma surrounding exposing menstrual status, an example of knowledge existing but stigma more strongly influencing a behaviour.

Beyond embarrassment, fear of being seen disposing or washing was a recurring theme [34,35,40,59,61,68,69,78,80,81,83,87,91]. This related to favouring methods of disposal or washing that were deemed secretive or hidden in order to limit their menstrual status being exposed [34,47,60,68,78,91]. Within this theme, menstruators noted they wanted to hide their used materials so that animals, specifically dogs, couldn't find their materials and expose their status [20,41,60,70,82,83,87]. They especially wanted to hide their status from males, specifically fathers and fathers-in-law [40,47,60,61,68,80]. This was in part due to fears of unwanted sexual advance [35], abuse and violence [19,69,81] and being forced to leave school at menarche so as to get married [59,68]. Other fears to use disposal methods stemmed from young menstruators being afraid that having a period was a "punishment from God," so hiding materials so that their parents would not find out [35], or that if seen, they would risk "infertility" or "being cursed" [87].

Cultural knowledge also influenced the choices that drove disposal and washing behaviours. For example, incinerators are often used for menstrual waste [35], and although they may be technically appropriate, in some cultures there are negative connotations surrounding the burning of menstrual blood [84] or instances where menstrual blood is kept to be used for ritual purposes [22,93]. These feelings were highlighted in Karibu et al's Nigerian study where "38.0% stated that they chose what they considered to be the best disposal method to ensure protection from metaphysical forces. . .[and]. . . 2.4% said they chose their method to avoid evil people" [55].

## Biased menstrual choices

When examining the language used in the papers, it became apparent that there is a bias in the way some disposal and washing methods, as well as menstrual materials themselves, were written about by the authors, most of whom are WASH researchers or practitioners. There has been a tendency to write about material use or disposal and washing behaviours with a view that some are superior to others, even where there is limited health or technical evidence to support this. Bias was highlighted in statements such as '*only* x % of interviewees use sanitary pads' (emphasis added, [42,90,91,94]). Although sometimes the bias was used in ways to

highlight unhygienic practices, for example "only a few used any kind of antiseptic soap or liquid [to wash their pads]" [24] it still singled out and shamed individuals rather than considering the myriad of factors contributing to their behaviours and the suitability of their washing and disposal practices to their personal circumstances. A similar bias occurs in Community Led Total Sanitation and some sanitation marketing programmes, where people are identified, and often shamed for their behaviours, regardless of their ability to change a situation, whether that be due to insufficient funds, lack of access to different options, or personal behavioural and cultural choices [103,104].

It must be understood that there are many reasons for choosing specific menstrual materials and disposal and washing behaviours, and that writing in a style that judges choices may vilify individuals, reinforce harmful taboos, and not succeed in changing behaviours to those which may improve the wellbeing of those who menstruate. WASH practitioners who read such studies may develop their own biases against behaviours which are in fact appropriate to local contexts and low risk to menstruators' health. For example, there has been a frequent bias in the WASH literature and programming against the use of 'cloths' or 'rags', often conflated as one. However, this does not consider that a cleaned and dried reusable pad or cloth may be just as hygienic and clean as a disposable pad, whilst a dirty rag poses obvious hygiene risks. In Chakravarty et al's paper, this infiltrated bias had directly affected material use, with one menstruator stating, "here we use only pads, we now [after MHH programming] realise how unhygienic it is to use cloth" [28]. There are several factors that contribute to menstrual material choice including, but not limited to, the preferences of and options for disposal and washing available to those who menstruate.

## Implications for WASH programming

When programming for MHH and the associated WASH facilities, WASH professionals must consider not just preferences for menstrual material choice, technologies, and disposal and washing practices, but also the physical and social perceptions of menstruators, the availability and form of knowledge in the cultural context, and the menstrual taboos and social stigma that continue to impact on those who menstruate across the globe. The focus of MHH programming must be to improve the physical, mental, and social well-being of menstruators within their own contexts (the very definition of menstrual health [3]), not to promote specific WASH facilities or technologies preferred by implementers; considering the drivers of menstrual behaviour throughout the lifecycle of materials will assist in improved experiences for menstruators.

## Limitations

This was a global review of published data. We originally aimed to highlight and understand the disposal and washing practices of those who menstruate around the world, with no geographical or economic limitations. However, of the 80 studies included in this review, only three were from high-income economies and another five were from upper-middle income economies. This was surprising given the media coverage in recent years surrounding the 'fatbergs' negatively impacting the functionality of sewers in cities in higher income countries [105]. Such fatbergs are attributed largely to the flushing of menstrual materials and wet wipes. Water utilities have put out calls to stop such behaviour [106], but, similar to the way users are often encouraged to change their menstrual material disposal behaviours in LMICs through the imparting of knowledge, these campaigns tend to assume that people flush these items because they do not know how harmful the practice is. Very limited research has investigated the behavioural drivers of menstrual material disposal in higher income economies.

Thus, although this study set out to reflect worldwide practices, due to the lack of data from higher income economies, this study is not representative of global practices. There were parallels within the small amount of high/upper-middle income economy data that showed similar themes and reasoning behind drivers of behaviour to lower-middle and low income economices, but due to the low proportion of these papers, we cannot determine with certainty whether these behaviours are universal. We join with Alda-Vidal et al. [15] in calling for further social science research on the clear gap of higher income economy data, and suggest it will be of particular interest in areas where fatbergs are impacting on WASH systems.

## Conclusion

This review demonstrates the complex nature of washing and disposal behaviours related to menstruation. Behaviours are often not solely reliant on one factor, but several interrelated considerations. It is the first review that has aimed at understanding why people choose to engage in various menstrual material disposal and washing practices.

WASH professionals and other implementers of menstrual disposal and washing facilities and services need to ensure that disposal and washing options are appropriate for their context. Even when facilities are installed and accessible, if considered inappropriate they will not be used [36,57,71]. In addition, educational policy needs to allow for the teaching of menstruation in a scientific, judgement-free zone, where those who menstruate feel comfortable to learn without the fear of embarrassment. Many young menstruators, and teachers in some instances, were highlighted as not having sufficient knowledge about menstruation and menstrual management, or were biased to certain materials and practices, with a specific emphasis on missing disposal information [21,24,30]. By creating a safe space to facilitate discussion, young menstruators will be able to learn more about this taboo topic.

Menstrual material disposal and washing is an area that is poorly understood globally, and as the first systematic review compiling these behaviours, this paper begins to provide clarity in an under-researched area. By exploring the drivers of disposal and washing behaviours, we demonstrate the interfaces between facilities, knowledge and taboos d. It is clearly important to integrate these aspects into the planning and provision of infrastructure systems that interface with MHH in order to provide accessible and appropriate facilities for all.

## Supporting information

**S1 Checklist. PRISMA 2009 checklist.**
(DOCX)

**S1 Table. Characteristics of studies included in the systematic review.**
(DOCX)

**S2 Table. Quality appraisal of included studies.**
(DOCX)

**S3 Table. Illustrative examples showing drivers of behaviours.**
(DOCX)

## Acknowledgments

The authors would like to thank Professor Jamie Bartram, Dr Celia Way, Dr Fiona Zakaria, Ms Mariam Zaqout and Mr Mofwe Kapulu for comments on an earlier version of this manuscript.

## Author Contributions

**Conceptualization:** Hannah Jayne Robinson, Dani Jennifer Barrington.

**Data curation:** Hannah Jayne Robinson, Dani Jennifer Barrington.

**Formal analysis:** Hannah Jayne Robinson.

**Investigation:** Hannah Jayne Robinson, Dani Jennifer Barrington.

**Methodology:** Hannah Jayne Robinson, Dani Jennifer Barrington.

**Project administration:** Hannah Jayne Robinson, Dani Jennifer Barrington.

**Supervision:** Dani Jennifer Barrington.

**Writing – original draft:** Hannah Jayne Robinson.

**Writing – review & editing:** Hannah Jayne Robinson, Dani Jennifer Barrington.

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
