## [Decision Letter · Decision Letter 0]

11 Jun 2021

PONE-D-21-01105

Drivers of menstrual material disposal and washing practices: A systematic review

PLOS ONE

Dear Dr. Robinson,

Thank you for submitting your manuscript to PLOS ONE. After careful consideration, we feel that it has merit but does not fully meet PLOS ONE’s publication criteria as it currently stands. Therefore, we invite you to submit a revised version of the manuscript that addresses the points raised during the review process.

Please submit your revised manuscript by 11July 2021. If you will need more time than this to complete your revisions, please reply to this message or contact the journal office at plosone@plos.org. Please include the following items when submitting your revised manuscript:

We look forward to receiving your revised manuscript.

Kind regards,

Balasubramani Ravindran, Ph.D

Academic Editor

PLOS ONE

Journal Requirements:

3. Please confirm that you have included all items recommended in the PRISMA checklist including:

-    the full electronic search strategy used to identify studies with all search terms and limits for at least one database.

-    an updated search and analysis that includes studies published since June 2019

-    See https://journals.plos.org/plosmedicine/article?id=10.1371/journal.pmed.1000100#pmed-1000100-t003 for guidance on reporting.

Thank you

5. Please include captions for your Supporting Information files at the end of your manuscript, and update any in-text citations to match accordingly. Please see our Supporting Information guidelines for more information: http://journals.plos.org/plosone/s/supporting-information

Reviewers' comments:

Reviewer's Responses to Questions

**Comments to the Author**

1. Is the manuscript technically sound, and do the data support the conclusions?

Reviewer #1: Yes

Reviewer #2: Yes

2. Has the statistical analysis been performed appropriately and rigorously? 

Reviewer #1: No

Reviewer #2: Yes

3. Have the authors made all data underlying the findings in their manuscript fully available?

Reviewer #1: Yes

Reviewer #2: Yes

4. Is the manuscript presented in an intelligible fashion and written in standard English?

Reviewer #1: Yes

Reviewer #2: Yes

5. Review Comments to the Author

Reviewer #1: The manuscript is well written .

But certain facts need to be cleared ,which is mentioned in the corrected manuscript.

The alignment of references could be checked.

New references could be further added.

Certain facts could be authenticated.

Reviewer #2: Review article is well written. References to be rechecked and written as per journals instruction.

References should be changed as per instruction of the journal.

Sufficient data enclosed and satisfactory

6. PLOS authors have the option to publish the peer review history of their article (what does this mean?). If published, this will include your full peer review and any attached files.

Reviewer #1: No

Reviewer #2: **Yes: **Dr. SumathiJones

---

## [Author Response · Author response to Decision Letter 0]

22 Sep 2021

Additional requirements:

• File names have been adjusted accordingly and formatting has been adjusted to ensure style compliance.

• No references have been retracted since initial submission.

• The search strategy is now included as Table 1.

• An updated search was performed that identified 2,310 additional papers (after duplicates were removed) in June 2021, this resulted in an additional 27 references being included. Analysis of the new references did not change the findings of this study.

• This manuscript utilised secondary data in the form of published, peer-reviewed journal articles and theses. All of these articles are available online, and DOIs are provided in the reference list where possible.

• Captions have been added, a new table has been included and existing tables have been updated accordingly.

Reviewer Comments:

1 – Alignment of all references has been addressed, and additional references have been added where requested.

2 - References have been shortened to use abbreviated titles, and DOIs have been included where possible.

---

## [Decision Letter · Decision Letter 1]

11 Nov 2021

Drivers of menstrual material disposal and washing practices: A systematic review

PONE-D-21-01105R1

Dear Dr. Robinson,

We’re pleased to inform you that your manuscript has been judged scientifically suitable for publication and will be formally accepted for publication once it meets all outstanding technical requirements.

Kind regards,

Balasubramani Ravindran, Ph.D

Academic Editor

PLOS ONE

Additional Editor Comments (optional):

Reviewers' comments:

Reviewer's Responses to Questions

**Comments to the Author**

1. If the authors have adequately addressed your comments raised in a previous round of review and you feel that this manuscript is now acceptable for publication, you may indicate that here to bypass the “Comments to the Author” section, enter your conflict of interest statement in the “Confidential to Editor” section, and submit your "Accept" recommendation.

Reviewer #1: All comments have been addressed

Reviewer #2: All comments have been addressed

2. Is the manuscript technically sound, and do the data support the conclusions?

Reviewer #1: Yes

Reviewer #2: Yes

3. Has the statistical analysis been performed appropriately and rigorously? 

Reviewer #1: N/A

Reviewer #2: Yes

4. Have the authors made all data underlying the findings in their manuscript fully available?

Reviewer #1: Yes

Reviewer #2: Yes

5. Is the manuscript presented in an intelligible fashion and written in standard English?

Reviewer #1: Yes

Reviewer #2: Yes

6. Review Comments to the Author

Reviewer #1: This review would facilitate the reader to understand the complex nature of washing and disposal behaviours related to

menstruation. It would educate the WASH professionals and educators too.It is a commendable work.

Reviewer #2: AUTHOR HAS ADDRESSED THE REVIEWERS COMMENTS

References are included

English and presentation is good

7. PLOS authors have the option to publish the peer review history of their article (what does this mean?). If published, this will include your full peer review and any attached files.

Reviewer #1: No

Reviewer #2: No

---

## [Editor Report · Acceptance letter]

25 Nov 2021

PONE-D-21-01105R1 

Drivers of menstrual material disposal and washing practices: A systematic review 

Dear Dr. Robinson:

I'm pleased to inform you that your manuscript has been deemed suitable for publication in PLOS ONE. Congratulations! Your manuscript is now with our production department. 

Kind regards, 

on behalf of

Dr. Balasubramani Ravindran 

Academic Editor

PLOS ONE